# Unlabeled Data Improves Fine-Grained Image Zero-shot Classification with Multimodal LLMs

**Yunqi Hong**[1], **Sohyun An**[1], **Andrew Bai**[1], **Neil Y. C. Lin**[2], **Cho-Jui Hsieh**[1]

[1]Computer Science Department, University of California, Los Angeles
[2]Mechanical and Aerospace Engineering Department, University of California, Los Angeles
{yunqihong, sohyun0423, andrewbai, chohsieh}@cs.ucla.edu, neillin@g.ucla.edu

## Abstract

Despite Multimodal Large Language Models (MLLMs) showing promising results on general zero-shot image classification tasks, fine-grained image classification remains challenging. It demands precise attention to subtle visual details to distinguish between visually similar subcategories—details that MLLMs may easily overlook without explicit guidance. To address this, we introduce AutoSEP, an iterative self-supervised prompt learning framework designed to enhance MLLM fine-grained classification capabilities in a fully unsupervised manner. Our core idea is to leverage unlabeled data to learn a description prompt that guides MLLMs in identifying crucial discriminative features within an image, and boosts classification accuracy. We developed an automatic self-enhancing prompt learning framework called AutoSEP to iteratively improve the description prompt using unlabeled data, based on instance-level classification scoring function. AutoSEP only requires black-box access to MLLMs, eliminating the need for any training or fine-tuning. We evaluate our approach on multiple fine-grained classification datasets. It consistently outperforms other unsupervised baselines, demonstrating the effectiveness of our self-supervised optimization framework. Notably, AutoSEP in average improves 13% over standard zero-shot classification and 3% over the best-performing baselines. Code is available at https://github.com/yq-hong/AutoSEP.

## 1   Introduction

While Multimodal Large Language Models (MLLMs) exhibit impressive zero-shot image classification capabilities on general image datasets [1, 2, 8, 20, 4, 15, 40], their performance often declines on fine-grained classification tasks. These tasks, which involve distinguishing between visually similar subcategories like specific bird, animal, or plant species [12, 10, 21], pose a challenge for MLLMs. The difficulty stems from the need to distinguish subtle visual differences; without explicit guidance, MLLMs tend to overlook crucial details or misinterpret key characteristics [35, 18, 22].

Previous efforts to enhance MLLMs' fine-grained classification abilities have largely relied on training the model with large-scale labeled datasets [12, 38], which are highly resource-intensive, demanding both significant computational resources and vast amounts of high-quality labeled data. This is especially challenging for fine-grained classification tasks, where collecting and annotating data is expensive and time-consuming due to the subtle visual distinctions between classes. In contrast, while unlabeled images are readily available at test time, there has been limited exploration into how such data could be leveraged to enhance classification performance. This motivates us to explore whether MLLMs can self-improve their fine-grained classification ability **using only unlabeled data and with only black-box access to the MLLM**.

But how can unlabeled data be leveraged to improve MLLMs' zero-shot prediction capabilities? While self-supervised learning literature [3, 13, 31] has demonstrated that unlabeled data can be used to

39th Conference on Neural Information Processing Systems (NeurIPS 2025).

train encoders to extract better discriminative features, it is nontrivial to achieve this in a zero-shot and black-box setting. To address this challenge, we develop Automatic Self-Enhancing Prompt Learning (AutoSEP), a novel framework for improving fine-grained visual classification MLLM with unlabeled data (see Figure 1). AutoSEP introduces an intermediate image description step, in which the MLLM is prompted to generate a textual description of an image's visual features. The MLLM then makes the final classification based on the original image and the additional textual description. This image description step can be viewed as mapping the raw image to text that highlights discriminative and relevant features for the task, analogous to the role of encoders in contrastive learning. To evaluate the quality of descriptions using unlabeled data, we design an instance-level classification scoring function that measures the MLLM's ability to retrieve the correct description for each image relative to those of other images. A prompt optimization algorithm is designed to iteratively improve the description-generation prompt based on this scoring function, without requiring white-box access to the MLLMs. By focusing on subtle yet critical visual differences, the MLLM iteratively improves its fine-grained classification performance without requiring any labeled training data.

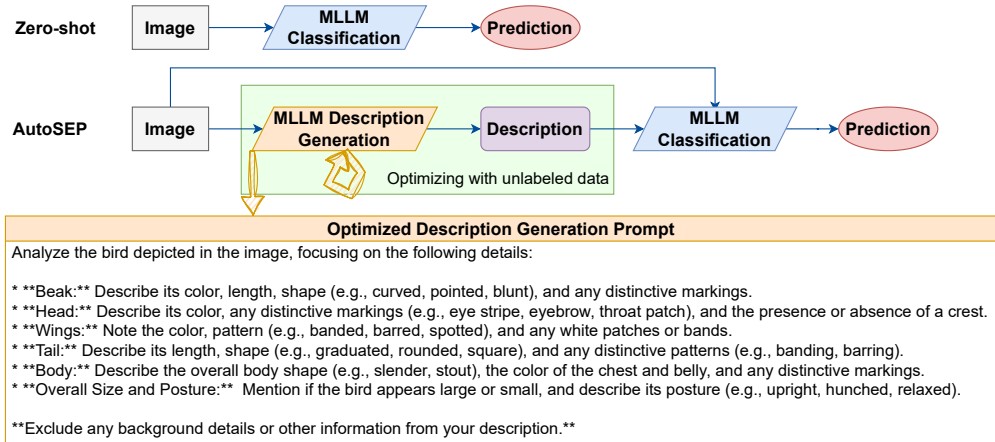

Figure 1: **Top:** Typical zero-shot pipeline for MLLM classification. **Middle:** Illustration of AutoSEP pipeline. **Bottom:** An example of an AutoSEP-optimized, description-generation prompt.

To summarize, our key contributions are as follows:

- We introduce a new self-supervised, black-box framework, AutoSEP, based on instance-level description retrieval to improve MLLMs fine-grained image classification capabilities.

- We show that AutoSEP significantly enhances MLLMs' fine-grained image classification capabilities (13% average improvement over zero-shot and 3% over the best-performing baselines) on 8 fine-grained tasks.

- To the best of our knowledge, this is the first work studying how to use unlabeled data to improve zero-shot image classification performance in the black-box setting for MLLMs.

## 2 Related Work

**Fine-grained image classification with MLLMs.** MLLMs have demonstrated remarkable performance across a broad spectrum of visual tasks, primarily due to pretraining on large-scale datasets [1, 20, 19, 40, 27, 14, 30]. However, since these models are predominantly trained on web-scraped corpora, they tend to underperform on fine-grained classification tasks [12, 10, 21]. To mitigate this limitation, prior works [12, 38] have employed extensive training using large-scale datasets. While effective, these approaches are computationally expensive and heavily reliant on high-quality labeled data, rendering them impractical in scenarios where fine-grained distinctions are critical. This motivates us to explore an alternative approach, which leverages readily available unlabeled images to improve fine-grained classification in MLLMs through a lightweight framework that requires neither labeled data nor white-box access to the model.

**Prompt optimization.** Automatic prompt optimization techniques have seen substantial progress in recent years, demonstrating their effectiveness in black-box, computationally constrained settings with a wide range of strategies for searching for effective prompts [26, 7, 37, 39, 24]. Specifically, several approaches [39, 24, 32] utilize LLMs to identify and summarize erroneous cases, which are then iteratively used to refine the prompts. Others [11, 9] adopt evolutionary strategies, generating diverse candidates through mutation and selection. Yang et al. [34] treats LLMs as optimizers, leveraging historical prompt trajectories and associated performance signals to guide prompt generation, while Yang et al. [36], Juneja et al. [17] decompose prompts into semantic segments for finer-grained control. Although these methods have shown strong performance, they largely depend on labeled data to supervise the optimization process. To mitigate this dependency, recent work such as SPO [33] formulates prompt optimization in a self-supervised manner by generating candidate prompts with LLMs and evaluating them through LLM-based pairwise preference comparisons of the resulting model outputs. In contrast, we proposed a higher-level framework for fine-grained visual classification that involves leveraging unlabeled images and explicit prompting for textual descriptions, whereas SPO is a lower-level prompt optimization technique. Furthermore, we find that SPO does not perform well even when incorporated into our framework due to its choice of scoring function (see Section 4).

## 3 Method

The key insight of AutoSEP is leveraging instance-level classification signal, which does not require label information, to inform the optimization of the description generation prompt for improving fine-grained classification. In Section 3.2, we introduce the technique of generating textual descriptions and combining it with image inputs to improve visual classification. In Section 3.3, we detail the procedures for learning the description-generation prompt in a self-supervised manner from instance-level classification scoring.

### 3.1 Problem Setting

Let $M$ denote the MLLM and $q$ denote the classification prompt. The vanilla zero-shot prediction is

$$\hat{y} = M(x, q), \tag{1}$$

where $x$ is the given image. While MLLMs often achieve strong performance on commonly seen basic-level classes, they tend to underperform on fine-grained classification tasks. We assume there exists a small unlabeled dataset $X = \{x_1, x_2, ..., x_n\}$ (more detailed experiments on different dataset sizes can be found in Section 4.3.) Our goal is to utilize $X$ to improve the zero-shot prediction accuracy on fine-grained categories. Also, we consider only **black-box access** to the MLLM, which means fine-tuning is prohibited.

### 3.2 Image Description Generation for Fine-grained Classification

Traditional self-supervised learning (SSL) leverages unlabeled data to contrastively learn better representations [3, 13, 31]. Inspired by SSL, we introduce an additional description generation step, where a "description generation prompt" $p$ is used to generate a text description $t$ for an image $x$. Both the description and the raw image are then jointly passed to the MLLM for predicting the label (see Figure 1):

$$\hat{y} = M(x, [q, t]), \quad \text{where } t = M(x, p). \tag{2}$$

Intuitively, $M(\cdot, p)$ functions as an encoder, translating the input $x$ into a text description $t$. This description then helps the MLLM conduct the final prediction.

However, a naive description generation prompt yields non-informative descriptions, and even worse, many descriptions might be misleading. Therefore, our method utilize unlabeled data $X$ to iteratively refine how MLLMs analyze and describe images, guiding them away from generic or incorrect visual features towards distinguishing and precise attributes.

### 3.3 Automatic Self-enhancing Prompt Learning

To learn a better description-generation prompt, we formalize an optimization objective based on a scoring function $\Psi(X, p)$ that evaluates the effectiveness of each candidate prompt $p$:

$$p^* = \arg\max_{p \in \mathcal{L}} \Psi(X, p), \tag{3}$$

where $\mathcal{L}$ represents the space of coherent natural language prompts.

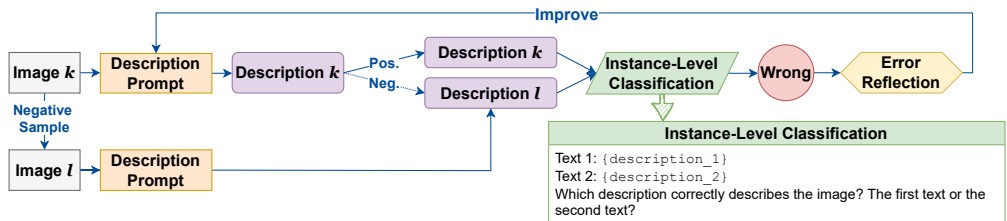

Figure 2: An illustration of automatic self-enhancing prompt learning with instance-level classification.

### 3.3.1 Instance-level Classification for Unsupervised Scoring

Inspired by traditional self-supervised learning methods [3, 13, 31], which learn the encoders to pull positive pairs closer and push negative pairs further apart contrastively, we propose to optimize the prompt so that the MLLM generates highly distinctive descriptions for each individual unlabeled image. By doing so, our method maximizes the inter-image distinctiveness of the generated descriptions.

We frame the optimization as an instance-level classification task, where the MLLM learns to correctly associate each image with its own generated description while distinguishing it from the descriptions of other images (See Figure 2). To quantify how well the MLLM can distinguish between different image descriptions, we define a sample matching correctness indicator $V(x_i, x_j)$. Given an image $x_i$ and two candidate descriptions $t_i$ and $t_j$, which are generated from images $x_i$ and $x_j$, respectively, we present the MLLM with a binary-choice prompt $G(t_a, t_b)$. The prompt asks "*Text 1: $\{t_a\}$. Text 2: $\{t_b\}$. Which description correctly describes the image? The first or the second?*", where $(t_a, t_b)$ is either $(t_i, t_j)$ or $(t_j, t_i)$, randomly shuffled with equal probability. The correctness indicator function is then formulated as:

$$V(x_i, x_j) := \begin{cases} \mathbb{1}[M(x_i, G(t_i, t_j)) = \text{``first''}], & Z = 0 \\ \mathbb{1}[M(x_i, G(t_j, t_i)) = \text{``second''}], & Z = 1, \end{cases} \quad (4)$$

where $Z \sim \text{Bernoulli}(0.5)$ determines the random ordering of descriptions to prevent positional bias. The function $V(x_i, x_j)$ returns 1 if the MLLM correctly selects the description corresponding to $x_i$, and 0 otherwise.

We compute the unsupervised scoring function that measures the model's ability to correctly match descriptions to their respective images. Formally, the scoring function is defined as:

$$\Psi(X, p) := \sum_{x_i \in X} \sum_{x_j \in X \setminus \{x_i\}} V(x_i, x_j). \quad (5)$$

To avoid iterating over all descriptions, $\Psi(X, p)$ can be approximated with random negative samples:

$$\hat{\Psi}(X, p, k) := \frac{1}{k \cdot |X|} \sum_{x_i \in X} \sum_{x_j \in \mathcal{O}_i} V(x_i, x_j), \quad (6)$$

where $\mathcal{O}_i \sim \text{RandomSample}(X \setminus \{x_i\}, k)$ represents a set of $k$ randomly sampled images from $X$, excluding $x_i$, acting as negative samples. $k \cdot |X|$ normalizes the score by the total number of evaluations conducted.

### 3.3.2 Description-generation Prompt Optimization

To begin the optimization process, we start with a simple yet effective initial generation prompt that instructs the MLLM to describe the object. The prompt can be found in Appendix A.

**Feedback and modification.** To iteratively refine the prompt, we adopt a feedback-driven approach that identifies and addresses the weaknesses in the current generation prompt. An illustration of AutoSEP workflow and prompts can be found in Appendix B. First, we collect error pairs

$J_{\text{error}} = \{(x_i, x_j) \mid x_i, x_j \in X, x_i \neq x_j, V(x_i, x_j) = 0\}$, where the generated descriptions failed to correctly distinguish between images. Then we sample a subset of $J_{\text{error}}$ and construct a diagnostic prompt to analyze potential shortcomings of the current prompt. Specifically, we present the MLLM with the current prompt together with examples of mismatched descriptions, and ask it to explain why the current prompt may have led to these errors. This process, referred to as the *Reflect* operator, $\text{Reflect}(\cdot)$, yields a set of potential reasons explaining the observed errors. Based on MLLM's feedback, we then prompt it to revise the current prompt by explicitly addressing the identified shortcomings. This refinement step is denoted as the *Modify* operator, $\text{Modify}(\cdot)$, which outputs an improved version of the generation prompt. Each instance of feedback produces a candidate revised prompt, resulting in multiple refined versions aimed at enhancing the distinctiveness of the generated descriptions. It has been shown in the literature that this reflect-and-modify approach is effective in automatic prompt engineering [24].

**Iterative optimization.** Our AutoSEP optimization process follows an iterative framework where the generation prompt is progressively refined to generate highly distinctive descriptions (Algorithm 1). Starting with an initial prompt, we evaluate its effectiveness by generating descriptions for a batch of images and forming an instance-level classification set by sampling negative images. Errors in the instance-level classification are identified and used to expand the prompt by adjusting it to better capture distinctive attributes. At each iteration, a set of candidate prompts is evaluated using a scoring function defined in Section 3.3.1, and only the top-performing prompts are retained for the next round. This iterative adaptation ensures that the MLLM increasingly focuses on fine-grained details, improving its ability to generate more informative descriptions. The process terminates after a predefined number of iterations $N$, and the best-performing prompt is selected as the final optimized prompt.

By maximizing $\hat{\Psi}(X, p, k)$, we iteratively refine the prompt $p$ to improve the distinctiveness of the generated descriptions, ultimately enhancing the MLLM's ability to perform fine-grained classification in a self-supervised manner.

---

**Algorithm 1** AutoSEP: Automatic Self-Enhancing Prompt Learning

---

**Require:** $p_0$: initial prompt, $N$: iterations, $X$: unlabeled dataset, $k$: negative samples per image, $b$: top prompts retained per iteration, $l$: number of reflections, $\hat{\Psi}(\cdot)$: scoring function

1: $P_0 \leftarrow \{p_0\}$      ▷ Initialize candidate prompt set
2: $\Gamma \leftarrow \{\gamma_0\}$, where $\gamma_0 = \{M(x_i, p_0) \mid x_i \in X\}$      ▷ Generate descriptions
3: **for** $t = 1$ to $N$ **do**
4:      $X_t \sim \mathcal{U}(X)$      ▷ Random minibatch from $X$
5:      $P_c \leftarrow P_{t-1}$
6:      **for** $p \in P_{t-1}$ **do**
7:          $J = \{(x_i, x_j) \mid x_i, x_j \in X_t, i \neq j\}$, where for each $x_i$, $k$ negative samples $x_j$ are randomly selected      ▷ Construct instance-level classification set
8:          $J_{\text{error}} = \{(x_i, x_j) \mid (x_i, x_j) \in J, V(x_i, x_j) = 0\}$      ▷ Collect errors
9:          $J_{\text{error}}^i \subset J_{\text{error}}$ is a sampled subset for each $i = 1, \ldots, l$
10:         $G = \{g_1, g_2, \ldots g_l\} = \bigcup_{i=1,\ldots l} \text{Reflect}(p, J_{\text{error}}^i)$      ▷ Reflect on the errors (Sec. 3.3.2)
11:         $H = \{h_1, h_2, \ldots h_l\} = \bigcup_{i=1,\ldots l} \text{Modify}(p, g_i, J_{\text{error}}^i)$      ▷ Modify prompts (Sec. 3.3.2)
12:         $P_c \leftarrow P_c \cup H$
13:         $\Gamma \leftarrow \Gamma \cup \{\{M(x_i, h) \mid x_i \in X\} \mid h \in H\}$      ▷ Generate descriptions for new prompts
14:      **end for**
15:      $S_c = \{\hat{\Psi}(X, p, k) \mid p \in P_c\}$      ▷ Evaluate prompts
16:      $P_t \leftarrow \{p \in P_c \mid \hat{\Psi}(X, p, k) \geq \tau\}$, where $\tau$ is the $b^{\text{th}}$ highest score in $S_c$
17: **end for**
18: **Return** $p^* \leftarrow \arg\max_{p \in P_N} \hat{\Psi}(X, p, k)$

---

## 4 Experiments

### 4.1 Experiment Settings

**Datasets.** We conduct experiments on four fine-grained image classification datasets, including CUB-200-2011 [29] (bird classification), iNaturalist 2021 [28] (various wild species), Stanford Dogs [6], and VegFru [25]. For each dataset, we construct fine-grained classification tasks on subsets

of categories that are particularly challenging for MLLMs. We construct three bird subsets with high confusion rates: *CUB_cuckoo* (Black-billed, Mangrove, and Yellow-billed cuckoo), *CUB_oriole* (Hooded, Orchard , and Scott's oriole), and *CUB_vireo* (Philadelphia, Red-eyed, and Warbling vireo). For iNaturalist, we select two subsets comprising visually similar species: *iNat_butterfly* (Symbrenthia lilaea, Claudina crescent, Elada checkerspot) and *iNat_lupine* (Arctic lupine, Silvery lupine, Arizona lupine). For Stanford Dogs, we focus on three closely related terriers: *StanfordDogs_terrier* (Lakeland terrier, Norwich terrier, Cairn terrier). For the VegFru dataset, we also select two subsets consisting of visually similar species: *VegFru_greens* (Dandelion, Shepherd's purse, Prickly lettuce) and *VegFru_allium* (Leek, Green Chinese onion, Bunching onion). All the results reported are evaluated on datasets that are balanced in class distribution.

**Baselines.** We compare our method with both optimization-free and optimization-based approaches. For optimization-free methods, we consider: (1) **Vanilla zero-shot**, where the MLLM is directly prompted without additional context; (2) **Zero-shot with descriptions**, using image descriptions generated from an initial human-crafted prompt; (3) **Zero-shot with majority vote**, where we prompt the MLLM to generate multiple responses with high temperature and take the most frequent prediction as the final answer; (4) **Few-shot with random labels**, where the MLLM is shown $m$ images with randomly assigned labels and then asked to classify the $(m + 1)^{th}$ image; (5) **Few-shot with MLLM labels**, where the MLLM is shown $m$ images with its own predicted labels and then asked to classify the $(m + 1)^{th}$ image; (6) **Multiple images display**, where the MLLM is directly shown $m$ unlabeled images before predicting the $(m + 1)^{th}$ image, as proposed by Jiang et al. [16]; (7) **K-means clustering**, where we cluster image features extracted by a pretrained image encoder, and then within each cluster, use the most frequent zero-shot prediction as the final label for all images in that cluster.

For prompt optimization-based methods, we consider three approaches to adapt them to the unsupervised setting: (1) **Optimization with random labels**, where we randomly assign labels to the images and use the classification accuracy under these random labels as the optimization objective; (2) **Optimization with majority vote**, where we iteratively assign pseudo-labels to images based on majority voting over multiple high-temperature outputs from the MLLM, and use the classification accuracy with respect to these pseudo-labels as the optimization objective; (3) **Self-Supervised Prompt Optimization (SPO)** [33], originally designed for text-only tasks, which perform pairwise comparisons with outputs generated by different prompts to infer their relative quality as signals to guide the optimization. We adopt SPO to our multimodal setting by asking the MLLM to compare the quality of image descriptions generated by different prompts. Following the original setup, we use GPT-4o [14] to conduct these comparisons. Prompts used for these baseline methods can be found in Appendix E.

**Models.** We conduct experiments with three state-of-the-art MLLMs: Gemini 1.5 Flash [27], GPT-4o [14], and Qwen2-VL-72B-Instruct [30], covering both proprietary and open-source models. We performed the experiments on local servers with 64 CPU cores and 4 Nvidia A6000 GPUs.

## 4.2 Experiment Results

Experiment results in Table 1 demonstrate that our method consistently outperforms all baseline approaches across all tested MLLMs. Notably, it achieves an average improvement of 13% over standard zero-shot classification and 3% over the best-performing baselines. These consistent gains highlight the robustness and generalizability of our approach across different model architectures and capabilities. Furthermore, our method operates without requiring any labeled data, making it particularly well-suited for low-resource or open-world scenarios.

**Description-generation prompt benefits from optimization.** We also observe that in some cases, the initial description generation prompt can slightly improve classification performance. However, these improvements are generally marginal. And in some cases, generating descriptions even degrades the performance. This suggests that unguided or naive prompting alone for description generation is insufficient. More principled guidance and optimization are necessary for the model to generate informative and task-relevant descriptions that more effectively support classification.

Table 1: Main results (accuracy %). AutoSEP outperforms baselines consistently across all tasks and MLLMs. Source of variability stems from samples in the evaluation dataset. Gemini refers to Gemini 1.5 Flash.

| Model | Gemini | GPT-4o | Qwen2-VL | Gemini | GPT-4o | Qwen2-VL |
|---|---|---|---|---|---|---|
| | *CUB_cuckoo* | | | *CUB_oriole* | | |
| ***Optimization-free*** | | | | | | |
| Vanilla zero-shot | $51.68_{\pm1.79}$ | $61.46_{\pm1.24}$ | $58.54_{\pm4.69}$ | $52.13_{\pm2.90}$ | $74.44_{\pm0.93}$ | $43.60_{\pm2.96}$ |
| Zero-shot with descriptions | $52.20_{\pm2.49}$ | $\underline{68.29}_{\pm1.00}$ | $51.46_{\pm2.60}$ | $46.74_{\pm2.31}$ | $73.71_{\pm1.52}$ | $50.79_{\pm3.92}$ |
| Zero-shot with majority vote | $51.22_{\pm2.04}$ | $61.95_{\pm1.95}$ | $59.02_{\pm1.98}$ | $54.38_{\pm0.55}$ | $75.28_{\pm2.01}$ | $46.74_{\pm4.95}$ |
| Few-shot with random labels | $38.05_{\pm6.14}$ | $43.90_{\pm7.48}$ | $46.83_{\pm9.74}$ | $40.05_{\pm12.1}$ | $56.85_{\pm12.3}$ | $36.63_{\pm6.50}$ |
| Few-shot with MLLM labels | $53.41_{\pm5.31}$ | $61.22_{\pm2.82}$ | $52.93_{\pm8.57}$ | $57.75_{\pm4.42}$ | $64.94_{\pm7.96}$ | $36.85_{\pm4.79}$ |
| Multiple images display [16] | $48.05_{\pm6.84}$ | $52.68_{\pm9.27}$ | $\textbf{65.12}_{\pm3.05}$ | $\textbf{59.78}_{\pm6.60}$ | $71.46_{\pm8.75}$ | $46.74_{\pm3.80}$ |
| K-means clustering | $53.17_{\pm15.2}$ | $59.51_{\pm15.0}$ | $56.59_{\pm2.26}$ | $51.46_{\pm5.83}$ | $67.19_{\pm8.11}$ | $44.94_{\pm7.07}$ |
| ***Optimization-based*** | | | | | | |
| With random labels | $\underline{59.15}_{\pm1.83}$ | $65.85_{\pm0.77}$ | $52.20_{\pm3.65}$ | $51.69_{\pm2.36}$ | $72.58_{\pm1.83}$ | $\underline{51.01}_{\pm1.83}$ |
| With majority vote | $44.88_{\pm4.69}$ | $67.32_{\pm2.10}$ | $53.41_{\pm2.36}$ | $57.90_{\pm1.56}$ | $73.03_{\pm1.74}$ | $38.65_{\pm1.74}$ |
| SPO | $55.98_{\pm1.34}$ | $66.59_{\pm2.63}$ | $56.10_{\pm6.19}$ | $49.49_{\pm2.71}$ | $\textbf{76.12}_{\pm2.16}$ | $46.52_{\pm3.37}$ |
| AutoSEP (Ours) | $\textbf{61.22}_{\pm2.30}$ | $\textbf{75.12}_{\pm1.83}$ | $\underline{63.41}_{\pm2.77}$ | $\underline{58.20}_{\pm0.88}$ | $75.96_{\pm0.90}$ | $\textbf{55.13}_{\pm3.58}$ |
| | *CUB_vireo* | | | *iNat_butterfly* | | |
| ***Optimization-free*** | | | | | | |
| Vanilla zero-shot | $49.21_{\pm3.36}$ | $75.73_{\pm1.83}$ | $55.28_{\pm3.05}$ | $61.14_{\pm2.11}$ | $68.00_{\pm1.46}$ | $40.25_{\pm4.65}$ |
| Zero-shot with descriptions | $51.46_{\pm4.11}$ | $86.07_{\pm0.90}$ | $50.79_{\pm3.72}$ | $56.57_{\pm3.45}$ | $71.14_{\pm3.88}$ | $48.25_{\pm2.48}$ |
| Zero-shot with majority vote | $49.21_{\pm1.80}$ | $76.69_{\pm2.01}$ | $55.06_{\pm3.01}$ | $62.00_{\pm2.14}$ | $66.07_{\pm1.86}$ | $46.56_{\pm2.59}$ |
| Few-shot with random labels | $45.62_{\pm8.03}$ | $57.08_{\pm18.2}$ | $43.82_{\pm7.78}$ | $45.43_{\pm13.8}$ | $64.00_{\pm10.7}$ | $32.13_{\pm8.47}$ |
| Few-shot with MLLM labels | $44.94_{\pm8.56}$ | $72.36_{\pm10.5}$ | $46.97_{\pm7.63}$ | $51.43_{\pm3.26}$ | $\underline{82.57}_{\pm14.3}$ | $42.13_{\pm12.2}$ |
| Multiple images display [16] | $\underline{56.18}_{\pm7.95}$ | $68.54_{\pm13.6}$ | $63.82_{\pm5.34}$ | $55.71_{\pm1.56}$ | $72.29_{\pm12.2}$ | $32.25_{\pm3.44}$ |
| K-means clustering | $40.22_{\pm9.25}$ | $57.08_{\pm11.2}$ | $57.53_{\pm15.0}$ | $58.57_{\pm5.89}$ | $59.14_{\pm5.39}$ | $38.38_{\pm6.92}$ |
| ***Optimization-based*** | | | | | | |
| With random labels | $41.24_{\pm6.85}$ | $72.81_{\pm3.13}$ | $48.31_{\pm4.44}$ | $53.24_{\pm1.72}$ | $79.86_{\pm4.43}$ | $45.13_{\pm1.71}$ |
| With majority vote | $53.71_{\pm5.41}$ | $\underline{87.19}_{\pm1.83}$ | $51.24_{\pm2.72}$ | $\underline{62.29}_{\pm1.63}$ | $61.07_{\pm1.89}$ | $\underline{51.50}_{\pm3.36}$ |
| SPO | $52.81_{\pm9.59}$ | $70.34_{\pm3.73}$ | $51.69_{\pm1.59}$ | $53.14_{\pm2.24}$ | $79.71_{\pm9.14}$ | $45.26_{\pm2.47}$ |
| AutoSEP (Ours) | $\textbf{59.18}_{\pm0.53}$ | $\textbf{87.42}_{\pm1.10}$ | $\textbf{65.45}_{\pm1.24}$ | $\textbf{66.57}_{\pm1.42}$ | $\textbf{82.71}_{\pm2.43}$ | $\textbf{55.38}_{\pm1.00}$ |
| | *iNat_lupine* | | | *StanfordDogs_terrier* | | |
| ***Optimization-free*** | | | | | | |
| Vanilla zero-shot | $62.00_{\pm2.72}$ | $70.00_{\pm3.59}$ | $44.22_{\pm3.74}$ | $60.80_{\pm2.13}$ | $91.39_{\pm0.48}$ | $73.20_{\pm2.40}$ |
| Zero-shot with descriptions | $56.67_{\pm1.92}$ | $73.00_{\pm1.25}$ | $44.00_{\pm2.49}$ | $66.22_{\pm2.41}$ | $\underline{91.42}_{\pm1.17}$ | $76.00_{\pm2.53}$ |
| Zero-shot with majority vote | $61.78_{\pm1.24}$ | $72.33_{\pm1.70}$ | $47.33_{\pm2.06}$ | $62.67_{\pm0.73}$ | $90.89_{\pm1.09}$ | $76.27_{\pm2.13}$ |
| Few-shot with random labels | $47.56_{\pm7.31}$ | $48.33_{\pm5.48}$ | $42.00_{\pm7.38}$ | $60.27_{\pm10.3}$ | $66.89_{\pm24.7}$ | $52.13_{\pm8.90}$ |
| Few-shot with MLLM labels | $60.22_{\pm11.4}$ | $69.33_{\pm3.89}$ | $46.44_{\pm6.75}$ | $66.67_{\pm10.3}$ | $87.56_{\pm3.47}$ | $64.67_{\pm6.84}$ |
| Multiple images display [16] | $\underline{63.47}_{\pm7.71}$ | $74.00_{\pm5.54}$ | $\underline{50.44}_{\pm2.86}$ | $65.87_{\pm4.04}$ | $79.78_{\pm9.04}$ | $77.73_{\pm4.43}$ |
| K-means clustering | $47.11_{\pm6.50}$ | $50.00_{\pm2.11}$ | $30.00_{\pm1.99}$ | $39.60_{\pm7.50}$ | $68.67_{\pm1.09}$ | $64.13_{\pm5.66}$ |
| ***Optimization-based*** | | | | | | |
| With random labels | $58.44_{\pm4.80}$ | $73.67_{\pm1.94}$ | $46.00_{\pm3.62}$ | $53.33_{\pm1.94}$ | $90.00_{\pm1.11}$ | $63.73_{\pm1.87}$ |
| With majority vote | $59.33_{\pm1.81}$ | $\underline{74.17}_{\pm1.83}$ | $47.33_{\pm4.37}$ | $62.27_{\pm1.82}$ | $91.11_{\pm1.41}$ | $\underline{77.74}_{\pm1.07}$ |
| SPO | $56.44_{\pm3.17}$ | $64.88_{\pm6.12}$ | $46.56_{\pm3.44}$ | $65.16_{\pm6.10}$ | $81.47_{\pm2.12}$ | $64.27_{\pm2.00}$ |
| AutoSEP (Ours) | $\textbf{66.15}_{\pm1.47}$ | $\textbf{76.60}_{\pm1.40}$ | $\textbf{54.67}_{\pm3.33}$ | $\textbf{69.28}_{\pm0.97}$ | $\textbf{92.18}_{\pm1.09}$ | $\textbf{81.70}_{\pm1.20}$ |
| | *VegFru_greens* | | | *VegFru_allium* | | |
| ***Optimization-free*** | | | | | | |
| Vanilla zero-shot | $\underline{86.00}_{\pm0.56}$ | $78.82_{\pm0.54}$ | $68.89_{\pm3.06}$ | $72.67_{\pm3.18}$ | $60.44_{\pm1.66}$ | $54.22_{\pm3.40}$ |
| Zero-shot with descriptions | $77.11_{\pm3.19}$ | $77.33_{\pm1.94}$ | $63.61_{\pm1.82}$ | $74.44_{\pm1.98}$ | $64.89_{\pm1.51}$ | $59.17_{\pm1.98}$ |
| Zero-shot with majority vote | $85.83_{\pm0.48}$ | $78.00_{\pm0.44}$ | $\underline{71.39}_{\pm2.53}$ | $75.56_{\pm1.82}$ | $62.44_{\pm2.57}$ | $56.89_{\pm2.37}$ |
| Few-shot with random labels | $74.44_{\pm6.30}$ | $70.22_{\pm8.47}$ | $33.33_{\pm0.00}$ | $67.33_{\pm3.18}$ | $61.33_{\pm10.6}$ | $33.33_{\pm0.00}$ |
| Few-shot with MLLM labels | $84.44_{\pm7.24}$ | $78.89_{\pm7.13}$ | $68.44_{\pm5.14}$ | $75.11_{\pm7.39}$ | $\textbf{73.11}_{\pm9.23}$ | $59.11_{\pm15.1}$ |
| Multiple images display [16] | $82.44_{\pm4.30}$ | $72.89_{\pm5.14}$ | $70.22_{\pm10.6}$ | $\textbf{79.56}_{\pm2.69}$ | $66.00_{\pm7.29}$ | $59.56_{\pm4.25}$ |
| K-means clustering | $67.78_{\pm5.24}$ | $67.56_{\pm4.94}$ | $64.00_{\pm5.19}$ | $51.33_{\pm14.1}$ | $50.89_{\pm9.33}$ | $42.67_{\pm7.72}$ |
| ***Optimization-based*** | | | | | | |
| With random labels | $80.56_{\pm4.33}$ | $\underline{79.78}_{\pm0.44}$ | $48.52_{\pm2.77}$ | $71.56_{\pm2.06}$ | $64.22_{\pm1.47}$ | $55.19_{\pm4.48}$ |
| With majority vote | $81.19_{\pm4.27}$ | $74.00_{\pm0.89}$ | $55.56_{\pm0.91}$ | $74.22_{\pm0.44}$ | $64.67_{\pm0.83}$ | $56.94_{\pm1.21}$ |
| SPO | $84.37_{\pm5.26}$ | $78.40_{\pm0.87}$ | $57.78_{\pm3.14}$ | $73.78_{\pm1.81}$ | $65.56_{\pm2.90}$ | $\underline{60.00}_{\pm0.91}$ |
| AutoSEP (Ours) | $\textbf{87.78}_{\pm0.70}$ | $\textbf{81.11}_{\pm0.70}$ | $\textbf{72.22}_{\pm0.91}$ | $\underline{78.89}_{\pm0.99}$ | $\underline{69.11}_{\pm0.83}$ | $\textbf{62.44}_{\pm2.44}$ |

**Instance-level classification is an effective unsupervised scoring signal, outperforming alternative unsupervised scoring strategies.** A key component of our method is the use of instance-level classification accuracy as the unsupervised scoring signal during optimization. To better understand its effectiveness, we compare it against three alternative unsupervised scoring methods for optimization: (1) Classification accuracy with randomly assigned labels; (2) Classification accuracy with majority vote pseudo-labels, where pseudo-labels are obtained by aggregating high-temperature predictions from the MLLM; (3) Pairwise output evaluation in SPO [33], which relies on pairwise comparisons of model outputs to estimate relative prompt quality.

While these alternatives sometimes yield performance gains over optimization-free baselines, they exhibit limited reliability and overall lower performance compared to our instance-level classification approach. Majority vote pseudo-labeling tends to perform better than using random labels, but both methods fall short in stability and effectiveness. Although SPO has demonstrated strong performance on several text-only reasoning and question answering tasks, it does not transfer well to fine-grained image classification. The results suggest that while LLMs can effectively compare the quality of reasoning paths in purely textual contexts, MLLMs struggle to consistently evaluate the informativeness of image descriptions for fine-grained classification. As a result, such evaluations offer weak and often noisy feedback signals for optimization. These results highlight the importance of instance-level classification as a robust and effective unsupervised signal.

## 4.3 Discussion and Analysis

**Learning dynamics.** We study how the classification performance evolves throughout the optimization process and how many iterations are usually required for the algorithm to take effect. Full details of experimental settings are mentioned in Appendix C.1. The average results and their variances reported in Figure 3a and 3b are computed by averaging these classification performances across all three runs. The Max results are computed by taking the best performance among the four prompts at each iteration, then averaging across the three independent runs.

Both the instance-level (Figure 3a) and class-wise (Figure 3b) classification performance consistently improves with more optimization iterations. More demonstrations on other datasets can be found in Appendix C.1. The correlational evidence suggests that instance-level classification serves as an effective unsupervised optimization signal to guide the model toward better class-wise discrimination. These results validate our design of leveraging instance-level feedback to drive prompt optimization for fine-grained classification in the absence of labeled data.

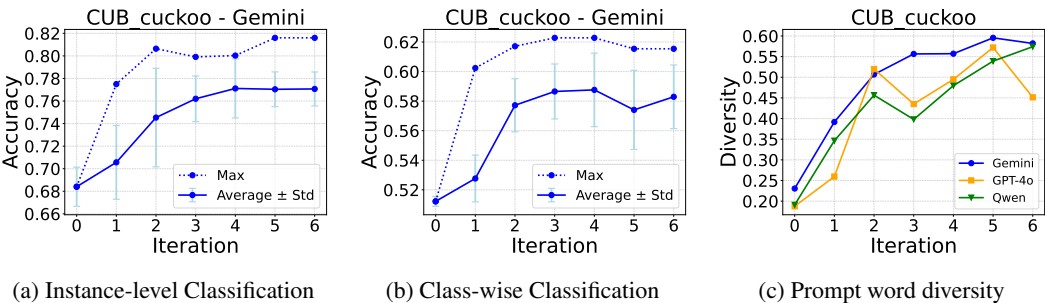

(a) Instance-level Classification   (b) Class-wise Classification   (c) Prompt word diversity

Figure 3: Evolution of metrics with increasing optimization iterations.

**Correlation between instance-level classification and class-wise classification.** To further support the effectiveness of instance-level classification as an optimization signal, we also compute the statistical Pearson correlation between instance-level and class-wise classification performance. Table 2 shows strong positive Pearson correlation across different settings, suggesting that improvements in instance-level accuracy are closely associated with better class-wise classification performance. This empirical evidence reinforces our hypothesis that instance-level supervision can serve as a powerful proxy to guide the description optimization for fine-grained image classification in the absence of labeled data.

**Diversity of generation prompts.** To analyze how much new information is introduced during the optimization process, we evaluate the diversity of the generated prompts using a quantitative diversity

Table 2: Correlation between instance-level classification and class-wise classification.

| Datasets | CUB_cuckoo | CUB_oriole | CUB_vireo | iNat_butterfly | iNat_lupine |
|---|---|---|---|---|---|
| Gemini 1.5 Flash | 0.72±0.11 | 0.47±0.24 | 0.70±0.19 | 0.63±0.22 | 0.53±0.02 |
| Qwen2-VL-72B-Instruct | 0.63±0.13 | 0.38±0.15 | 0.64±0.10 | 0.84±0.11 | 0.70±0.17 |

score. Specifically, for each prompt, we count the number of semantic keywords (lemmas of nouns, verbs, and adjectives excluding stop words), as well as the number of unique words, defined as words that appear only in that specific prompt and not in any other prompt throughout the optimization. These counts are normalized to compute the diversity score for each prompt. Detailed definition can be found in Appendix F.

As shown in Figure 3c, the diversity score steadily increases over iterations, indicating that the optimization process encourages the generation of increasingly varied and semantically rich prompts. This trend suggests that the model is progressively exploring a broader and more informative prompt space, which may potentially contribute to improved classification performance.

**Computational complexity.** Let $b$ denote the number of top prompts retained, $l$ is the number of prompt reflections generated per retained prompt, $k$ is the number of negative samples used per image in the instance-level classification, and $n$ is the number of unlabeled images. Each iteration of our algorithm involves generating $b \cdot l$ reflections, and $b \cdot l$ modified candidate prompts accordingly. For each of these prompts, we generate descriptions for all $n$ images, resulting in $b \cdot l \cdot n$ quiries to the MLLM for description generation. Additionally, instance-level classification requires comparing each image's description against $k$ negative samples under each prompt, leading to $b \cdot l \cdot k \cdot n$ further MLLM queries. Therefore, the total number of MLLM queries per iteration is $(2 + n + kn) \cdot bl$. In our implementation, we typically set $b = 4, l \leq 5$, and $k = 2$, resulting in an approximate query complexity of $\mathcal{O}(60n)$ per iteration.

For optimization-free baselines, the computational cost is relatively low, since they do not require iterative prompt refinement. For optimization-based baselines, we used the same experimental setup as AutoSEP, resulting in comparable computational complexity. More specifically, optimization with random labels takes approximately $\mathcal{O}(2(1 + n) \cdot bl) = \mathcal{O}(40n)$ MLLM queries per iteration, as it involves only a single classification per image. Optimization with majority vote takes approximately $\mathcal{O}(2(1 + n) \cdot bl + cn) = \mathcal{O}(45n)$, with $c = 5$ denoting the number of majority votes collected for updates in each iteration. SPO [33] is more lightweight, requiring only $\mathcal{O}(2n)$ queries per iteration, since it only compares with the previous prompt and does not rely on several candidate prompts.

The actual runtime depends on the inference speed of the MLLMs. For example, one optimization iteration on 60 images with Gemini 1.5 Flash takes 12 minutes in average, demonstrating that AutoSEP is computationally feasible for practical use.

**More samples, stronger instance-level signals, more stable optimization.** We investigate how the number of unlabeled samples used during optimization impacts the performance of our method. Figure 4 shows a clear trend that the classification performance improves with increased number of samples. More demonstrations on other datasets can be found in Appendix C.2.

When the sample size is too small, instance-level classification is constrained by limited variation and coverage, making it difficult to reliably evaluate a prompt's ability to generate distinctive and meaningful descriptions. This leads to suboptimal performance and introduces more noise and instability into the optimization process. In contrast, with more samples, the instance-level signal becomes more robust, enabling more stable optimization and resulting in consistent gains in class-wise classification accuracy. Notably, our method begins to perform reliably with **as few as 60 unlabeled samples**, demonstrating its practicality and scalability even in limited-data settings.

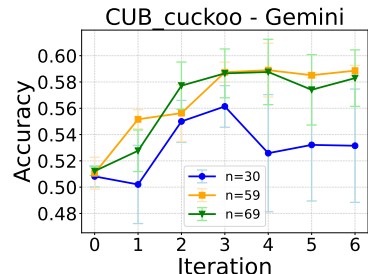

Figure 4: Classification accuracy of Gemini with various number of samples for optimization.

## 5 Conclusion

In this paper, we present a self-supervised prompt learning framework that improves the fine-grained image classification capabilities of MLLMs using only unlabeled data and black-box MLLM access. By optimizing the prompt for image description generation through instance-level classification feedback, our method enables MLLMs to focus on subtle, discriminative features critical for distinguishing between visually similar categories. Extensive experiments across multiple fine-grained classification tasks and MLLMs demonstrate the robustness of our approach.

**Limitations.** Our method relies on the implicit assumption that subtle, class-discriminative visual features can be accurately captured and described using natural language. This may not hold in cases where visual distinctions are extremely fine-grained or difficult to verbalize. Additionally, although our approach is fully unsupervised, it still requires a sufficient number of unlabeled samples to ensure stable and effective prompt optimization. The absence of labeled data inherently limits the performance compared to fully supervised methods that benefit from explicit class-level guidance.

## Acknowledgments and Disclosure of Funding

This work is supported by NSF 2048280, 2325121, 2244760, 2331966 and ONR N00014- 23-1-2300:P00001.

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

# A   Examples of Descriptions

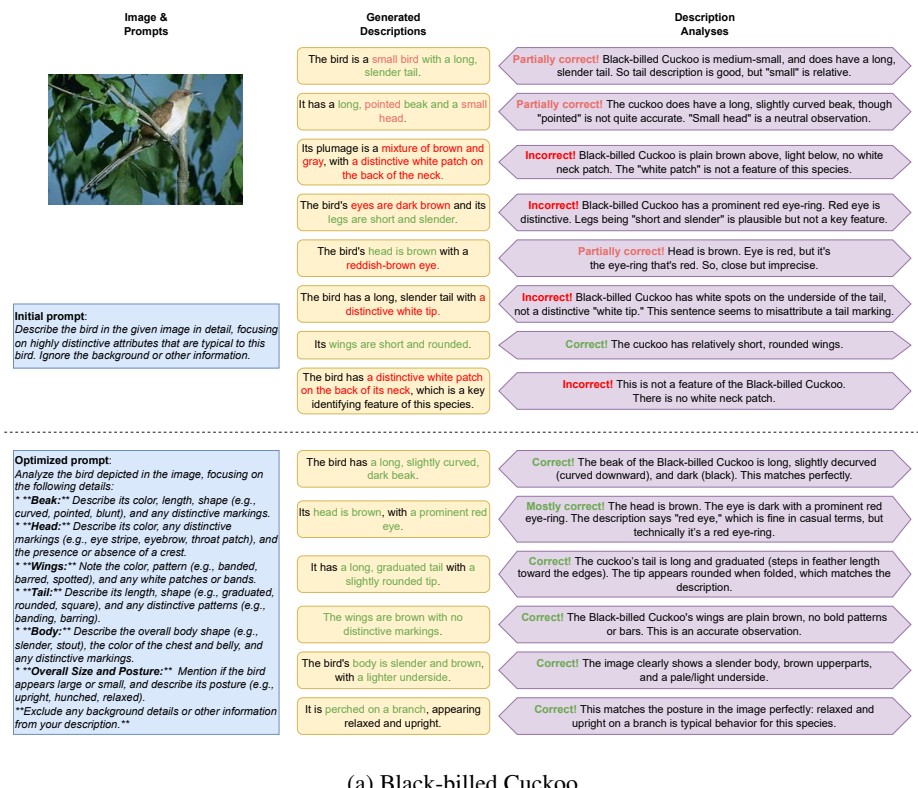

(a) Black-billed Cuckoo

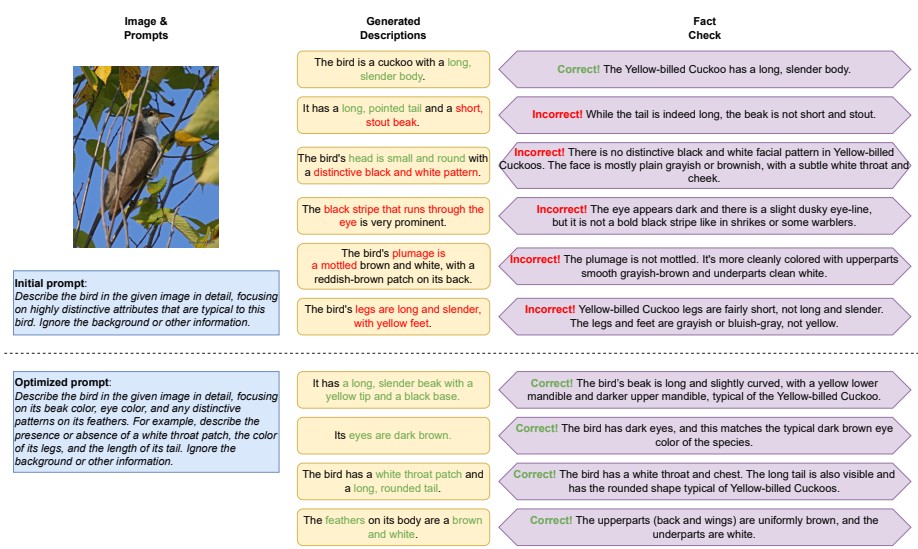

(b) Yellow-billed Cuckoo

Figure 5: Examples showcasing descriptions generated from two different prompts using Gemini 1.5 Flash. Attributes highlighted in green indicate correct information, while those in red denote incorrect attributes.

Figure 5 shows two examples of image descriptions generated by Gemini 1.5 Flash [27] using prompts before and after optimization. The initial descriptions tend to be vague or misleading, often misrepresenting key visual attributes. In contrast, the optimized prompts lead to more accurate and reliable descriptions that better reflect the distinctive features of the images.

# B  Detailed AutoSEP Workflow and Prompts

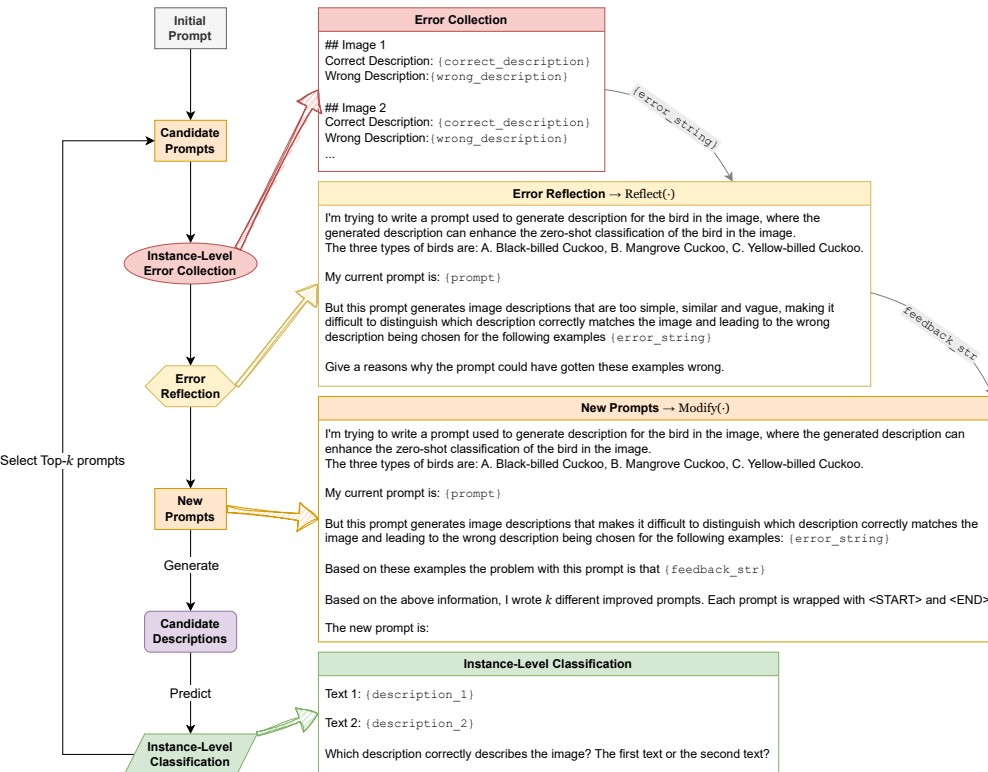

Figure 6: **Workflow** of AutoSEP with Optimization Prompts. Take *CUB_cuckoo* as an example.

Figure 6 shows the detailed workflow of AutoSEP and prompts used in the optimization. The prompt learning procedure is introduced in Section 3.3.2.

# C  Additional Ablation Studies

## C.1  Learning Dynamics

We study how the classification performance evolves throughout the optimization process and how many iterations are usually required for the algorithm to take effect. For each experimental setting, we perform three independent runs. In each run, we retain the top four candidate prompts at every iteration based on the scoring function. We use the average classification performance of these four prompts to represent the performance for that iteration. The average results and their variances reported in Figure 7 are computed by averaging these classification performances across all three runs. The Max results are computed by taking the best performance among the four prompts at each iteration, then averaging across the three independent runs.

## C.2  Number of Samples

More experiments on different datasets are conducted to study the effect of the number of samples on our AutoSEP framework (See Figure 8). The results are quite consistent across different classification tasks. When the sample size is too small, the optimization process is unstable with large variance. In contrast, with more samples, the instance-level signal becomes more robust, resulting in more stable optimization and consistent gains in class-wise classification accuracy.

## C.3  Different Sampling Methods

We conducted this ablation study to examine how the sampling strategy affects the final results. Specifically, we replaced the random image sampling in AutoSEP with two alternatives: sampling only same-class images

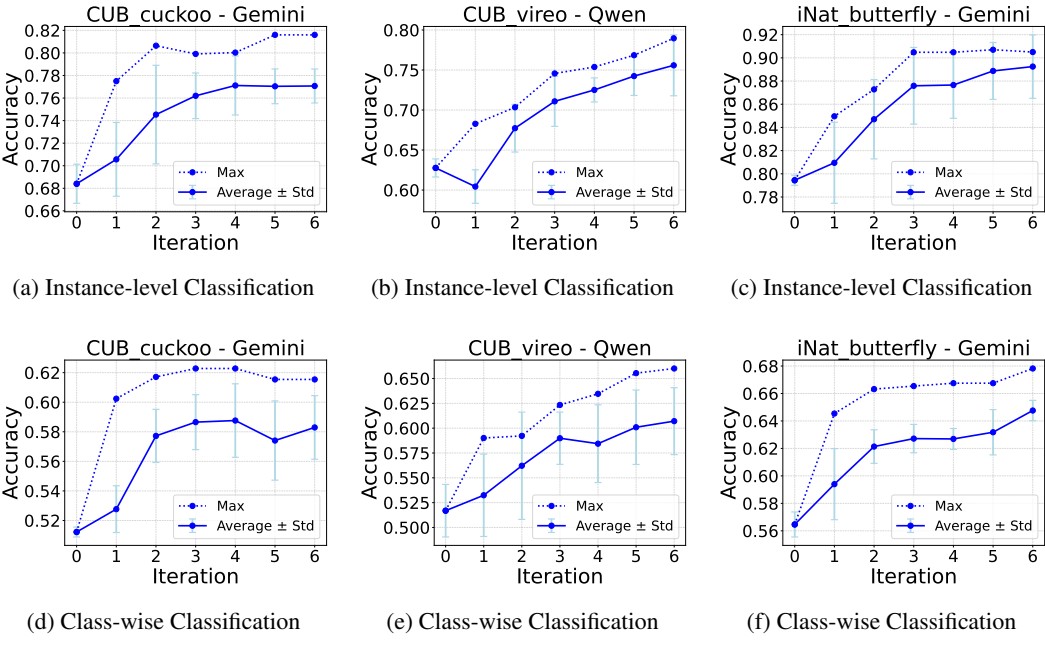

Figure 7: Classification accuracy as a function of iteration.

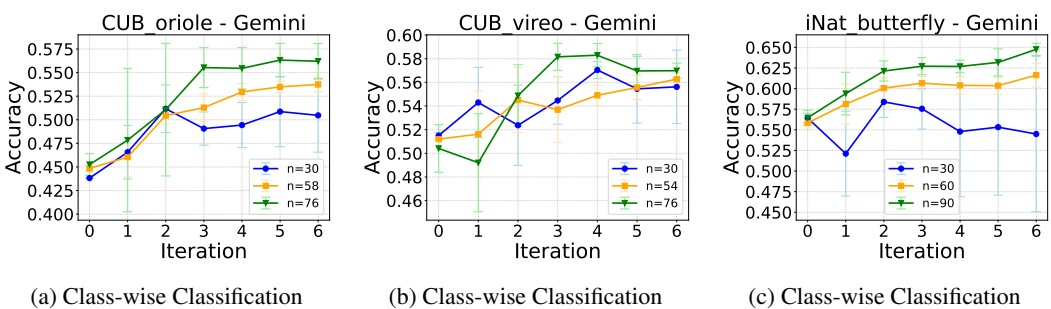

Figure 8: Classification accuracy of Gemini with various number of samples for optimization.

or only different-class images for instance-level classification. As shown in Table 3, neither same-class nor different-class sampling leads to a notable change in performance.

The goal of our description generation process is to elicit detailed, fine-grained understanding of each image, instead of enforcing class-level discrimination. This image-specific descriptive richness is then used to guide the classification process in a separate classification prompt. In this context, encountering visually similar images–even from the same class–is not a flaw but an opportunity: it encourages the model to uncover subtle, discriminative visual cues that may otherwise be overlooked. In traditional self-supervised learning methods, similar images from the same class are often sampled together to learn meaningful feature representations through contrastively learning. Our framework is inspired by this idea and pushes the model to verbalize these subtle differences, which ultimately improves the downstream classification.

Table 3: Results (accuracy %) of different sampling processes.

| Gemini 1.5 Flash | CUB_cuckoo |
|---|---|
| AutoSEP with different-class image sampling | $60.98_{\pm 3.59}$ |
| AutoSEP with same-class image sampling | $61.58_{\pm 3.11}$ |
| AutoSEP with random image sampling (ours) | $61.22_{\pm 2.30}$ |

# D  Additional Experiment Results

## D.1  Results with Gemini 2.0 Flash

We conducted additional experiments with Gemini 2.0 Flash [5], with the results presented in Table 4. While Gemini 2.0 Flash demonstrates a substantial improvement over Gemini 1.5 Flash, its performance does not surpass that of GPT-4o on our benchmark tasks.

Table 4: Results (accuracy %) on Gemini 2.0 Flash.

| Gemini 2.0 Flash | CUB_vireo | StanfordDogs_terrier |
|---|---|---|
| Vanilla zero-shot | $64.27_{\pm 1.31}$ | $74.60_{\pm 0.47}$ |
| Zero-shot with descriptions | $64.27_{\pm 1.65}$ | $74.12_{\pm 1.21}$ |
| AutoSEP (ours) | $\mathbf{69.05}_{\pm 1.78}$ | $\mathbf{85.94}_{\pm 1.05}$ |

Nevertheless, our AutoSEP method consistently improves the performance of Gemini 2.0 Flash across both datasets, further confirming its effectiveness across different MLLMs.

## D.2 Results with smaller MLLMs

We conducted additional experiments using Qwen2-VL-7B-Instruct [30]. Since AutoSEP relies on a strong agent for reflection and modification, we adopted a hybrid setup in which Gemini 1.5 Flash [27] served as the Reflect and Modify operators as well as for image description generation, while Qwen2-VL-7B-Instruct was used for instance-level classification and evaluation. This setup follows a common practice in prior prompt optimization studies [32, 33, 23]. As shown in Table 5, our method also yields substantial performance gains on Qwen2-VL-7B-Instruct, demonstrating its effectiveness on smaller models.

Table 5: Results (accuracy %) on Qwen2-VL-7B-Instruct.

| Qwen2-VL-7B-Instruct | CUB_cuckoo | CUB_oriole |
|---|---|---|
| Vanilla zero-shot | $40.24_{\pm 3.43}$ | $39.78_{\pm 7.41}$ |
| Zero-shot with descriptions | $39.27_{\pm 2.68}$ | $50.45_{\pm 1.56}$ |
| AutoSEP (ours) | $\mathbf{61.79}_{\pm 2.31}$ | $\mathbf{54.02}_{\pm 1.66}$ |

# E Prompt Details

In this section, we provide the prompts used for baselines in our experiments. Blue boxes show those used directly for classification; orange box shows the description generation prompt before optimization; red box shows the prompt used for SPO [33].

**Zero-shot (*CUB_cuckoo*)**

```
# Task
Determine what kind of bird this image shows from the following options:

A. Black-billed Cuckoo
B. Mangrove Cuckoo
C. Yellow-billed Cuckoo

Answer the letter from A to C as prediction.

The answer is:
```

**Zero-shot with image description (*CUB_cuckoo*)**

```
# Task
Determine what kind of bird this image shows from the following options:

A. Black-billed Cuckoo
B. Mangrove Cuckoo
C. Yellow-billed Cuckoo

Answer the letter from A to C as prediction.

# Prediction
Text: The image shows the following features: {description}
The answer is:
```

**Few-shot with random/MLLM labels (*CUB_cuckoo*)**

```
Your task is to classify the image to three birds: A. Black-billed Cuckoo, B. Mangrove Cuckoo, C.
↪   Yellow-billed Cuckoo.

The classification of the 1 image is: C
```

```
The classification of the 2 image is: B

The classification of the 3 image is: C
...

The classification of the last image is: (Answer Letter A or B or C)
```

**Multiple images display [16] (*CUB_cuckoo*)**

```
Given the first k image examples, answer the following question:

What kind of bird does the last image show: A. Black-billed Cuckoo, B. Mangrove Cuckoo, C.
↪  Yellow-billed Cuckoo.

Directly Answer Letter A or B or C without any further information
```

**Initial image description generation (*CUB_cuckoo*)**

```
# Task
Describe the bird in the given image in detail, focusing on highly distinctive attributes that
↪  are typical to this bird. Ignore the background or other information.
```

**SPO [33] description quality evaluation (*CUB_cuckoo*)**

```
I'm trying to write a prompt used to generate description for the bird in the image, where the
↪  generated description can enhance the zero-shot classification of the bird in the image.
The three types of birds are: A. Black-billed Cuckoo, B. Mangrove Cuckoo, C. Yellow-billed Cuckoo.

There are two descriptions of the given images generated by two different prompts:

Text 1: {description_1}

Text 2: {description_2}

Which description better describes the image? The first text or the second text? Provide your
↪  analysis and the choice you believe is better, using XML tags to encapsulate your response.

<analyse>Some analysis</analyse>
<choose>First/Second (the better answer in your opinion)</choose>
```

# F  Diversity Score

Quantifying the amount of new semantic content introduced during optimization is essential for understanding the evolution of prompts. To this end, we design a diversity score that captures two aspects:

- Semantic richness, measured by counting the number of unique lemmas of content words (nouns, verbs, adjectives) after stop-word removal, denoted as($T(p)$);
- Uniqueness across prompts, defined as the number of words that appear exclusively in a given prompt and not in any others, denoted as($U(p)$).

To balance the contributions of these components, we normalize each by the maximum value observed across all prompts in the optimization:

$$\tilde{T}(p) = \frac{T(p)}{\max_i T(p_i)}, \qquad \tilde{U}(p) = \frac{U(p)}{\max_i U(p_i)}. \tag{7}$$

The final diversity score is then computed as the sum of these normalized terms:

$$D(p) = \tilde{T}(p) + \tilde{U}(p). \tag{8}$$

This metric captures inter-prompt uniqueness, which is central to our analysis of how prompts diversify throughout the optimization process.

