# OpenReview forum: "Unlabeled Data Improves Fine-Grained Image Zero-shot Classification with Multimodal LLMs"
_NeurIPS.cc/2025/Conference — NeurIPS 2025 poster_

### Official Review · Reviewer_AiFG · 2025-06-28

**Clarity:** 3
**Significance:** 2
**Originality:** 3
**Rating:** 4
**Confidence:** 3

**Summary:**

For zero-shot fine-grained image classification with black-box multi-modal large language models (MLLM), this paper proposes to use an automatic prompt learning framework. It uses the MLLM to generate a description based on the image and a regular prompt and refines it with unlabeled data to generate a “distinctive descriptions for each unlabeled image”. It does this iteratively by, finally, giving the MLLM a binary choice prompt to select between two generated descriptions, one for the image in question and the other for another random image in the dataset.

**Questions:**

1.	Improve the comparison of results as described above.
2.	Justify having access to the parent class name and subset creation.
3.	Justify the query complexity.
I would be open to revising my rating if these concerns are addressed.

**Ethical Concerns:**

["NO or VERY MINOR ethics concerns only"]

**Final Justification:**

The rebuttal successfully addresses my comments and, as far as I can tell, those of the other reviewers as well.

**Limitations:**

MLLM should be knowledgeable of the attributes defining the fine-grained classes.
The paper clearly lists two other limitations.

**Paper Formatting Concerns:**

Sec. 3.3.2 refers the reader to Appendix D to find the “initial prompt”. However, Appendix A is more helpful in this regard.
Line 289 refers to Fig. 8 in the appendix rather than Fid. 4 in the main text.

**Quality:**

3

**Strengths And Weaknesses:**

Strengths:

1.	Getting an MLLM to generate prompts that increasingly focus on fine-grained details in this black-box MLLM fine-grained classification setting.
2.	The method’s ability to use instance level classification signal for unsupervised prompt optimization.
3.	The paper is, in general, clear.

Weaknesses:

1.	The initial prompts use the name of the parent class in the prompt, e.g., “the bird” in “Describe the bird in the given image in detail, focusing on highly distinctive attributes that are typical to this bird. Ignore the background or other information.”  While each dataset is limited to the parent class "bird," providing this information explicitly to an MLLM can be an unfair advantage.
2.	The number of times the MLLM gets queried is high. Initial prompt, iteration (e.g., 6), unlabeled image samples, prompts, prompts retained, reflection, modification, all involve prompting the MLLM. Computational complexity (query complexity), as described in lines 274 onwards, is high.
3.	The paper constructs fine-grained classification tasks on subsets of categories in these datasets. Provide justification (from literature or otherwise) for this. Provide results similar to, e.g., Saha et al., Improved Zero-Shot Classification by Adapting VLMs with Text Descriptions, CVPR 2024.
4.	The experiments have been done with large MLLMs. It would be beneficial to know the performance of smaller variants.
5.	The diversity score (Fig. 3c) is not clearly defined. Can’t the standard metrics be used?
6.	A major concern is the absence of comparison of results with existing work. The comparisons on variants of experiments (although logical) are crafted in this paper itself.

---

> ### Author Rebuttal · Authors · 2025-07-30
>
> We’re very thankful to the reviewer’s detailed and valuable feedback. We answer all of the concerns and questions below.
>
> **1. Parent class name in initial prompt**
> > **Reviewer’s concern**: The initial prompts use the name of the parent class in the prompt, e.g., “the bird”. While each dataset is limited to the parent class, providing this information explicitly to an MLLM can be an unfair advantage.
>
> **Our Response**:
> We thank the reviewer for bringing up this concern. To evaluate the impact of including the parent class name in the initial prompt, we replaced “the bird” with a more generic term, “the object,” and re-ran the experiments. The results are presented in the Table below.
>
> **Table: Results of different initial prompts.**
> | CUB_cuckoo | Gemini 1.5 Flash | Qwen2-VL-72B-Instruct |
> |:- |:-:|:-:|
> | AutoSEP with “the object” in initial prompt | 60.82 ± 3.43 | 63.72 ± 1.01  |
> | AutoSEP with “the bird” in initial prompt (ours)| 61.22 ± 2.30 | 63.41 ± 2.77  |
>
> We observe that this change has minimal effect on the final performance, suggesting that explicitly including the parent class name in the initial prompt does not provide a significant advantage in our setting.
>
> ****
> **2. Computational cost**
> > **Reviewer’s concern**: The number of times the MLLM gets queried is high. Computational complexity (query complexity) is high.
>
> **Our Response**:
> While our method involves multiple MLLM queries during the prompt optimization phase, this process is performed only once offline. 12 minutes per iteration with 60 examples using Gemini 1.5 Flash is **a modest runtime given the consistent and substantial performance gains** across diverse fine-grained classification tasks. Our method is practical, scalable, and requires no model fine-tuning or internal access, making it highly accessible and model-agnostic. It is also data-efficient, relying only on a small number of unlabeled examples. Compared to methods that require retraining, our approach offers a strong balance of efficiency and effectiveness, with minimal runtime overhead during inference (one forward pass per image for description generation and one for prediction).
>
> ****
> **3. Data subset creation**
> > **Reviewer’s concern**: The paper constructs fine-grained classification tasks on subsets of categories in these datasets. Provide justification (from literature or otherwise) for this. Provide results similar to, e.g., Saha et al., Improved Zero-Shot Classification by Adapting VLMs with Text Descriptions.
>
> **Our Response**:
> We thank the reviewer for raising this important point. Our decision to construct fine-grained classification tasks using subsets of categories is motivated by both practical considerations in the MLLM setting and some prior work.
>
> In these fine-grained datasets, there’re over a hundred categories. Unlike CLIP-style models, which computes similarity scores across hundreds of categories to make predictions, asking an MLLM to choose from over a hundred fine-grained labels in a single prompt is rarely done and less reliable with the results. Instead, we focus on **challenging subsets of categories where zero-shot performance is low**, making them meaningful benchmarks for evaluating improvements. This allows us to study performance gains in hard fine-grained settings, rather than on categories that are easily separable by existing MLLMs.
>
> This design is **consistent with prior work**. For example,
> * LLM-Mutate[1] used 6 species of Lichen (fungi), 5 species of Wrasse (fish) for classification.
> * Finedefics[2] constructed their dataset using 4 candidate options per image to test fine-grained classification performance.
>
> **Reference:**
>
> [1] Chiquier, Mia, Utkarsh Mall, and Carl Vondrick. "Evolving interpretable visual classifiers with large language models." European Conference on Computer Vision. Cham: Springer Nature Switzerland, 2024.
>
> [2] He, Hulingxiao, et al. "Analyzing and boosting the power of fine-grained visual recognition for multi-modal large language models." arXiv preprint arXiv:2501.15140 (2025).
>
> ****
> **4. Results on smaller models.**
> > **Reviewer’s concern**: The experiments have been done with large MLLMs. It would be beneficial to know the performance of smaller variants.
>
> **Our Response**:
> We appreciate this insightful suggestion to explore on smaller models. We conducted additional experiments using **Qwen2-VL-7B-Instruct**. Since automatic prompt optimization requires stronger agents for reflection and modification, we explored a hybrid setup where Gemini 1.5 Flash was used as the Reflect and Modify operators and image description generation, while the Qwen 7B model was used for instance-level classification and evaluation. This is a common setting to use a stronger model for reflection[3][4][5]. The results below indicate that our method also leads to **significant improvements** on Qwen2-VL-7B-Instruct.
>
> **Table: Results on Qwen2-VL-7B-Instruct.**
> | CUB_cuckoo | Qwen2-VL-7B-Instruct | CUB_oriole | Qwen2-VL-7B-Instruct |
> |:- |:-:|:-|:-:|
> | Vanilla zero-shot | 40.24 ± 3.43     |Vanilla zero-shot  | 39.78 ± 7.41  |
> | Zero-shot with descriptions| 39.27 ± 2.68 | Zero-shot with descriptions | 50.45 ± 1.56 |
> | AutoSEP (ours)  | **61.79** ± 2.31   |AutoSEP (ours)  | **54.02** ± 1.66 |
>
> These results demonstrate that our method can effectively enhance fine-grained classification performance even for smaller models when paired with a stronger MLLM as the optimization engine. We will include these results to our revised paper to strengthen the comprehensiveness of our work.
>
> **Reference:**
>
> [3] Wang, Xinyuan, et al. "Promptagent: Strategic planning with language models enables expert-level prompt optimization." arXiv preprint arXiv:2310.16427 (2023).
>
> [4] Xiang, Jinyu, et al. "Self-supervised prompt optimization." arXiv preprint arXiv:2502.06855 (2025).
>
> [5] Murthy, Rithesh, et al. "Promptomatix: An Automatic Prompt Optimization Framework for Large Language Models." arXiv preprint arXiv:2507.14241 (2025).
>
> ****
> **5. Regarding diversity compute**
> > **Reviewer’s concern**: The diversity score (Fig. 3c) is not clearly defined. Can’t the standard metrics be used?
>
> **Our Response**:
> We appreciate the reviewer’s feedback regarding the clarification of diversity score. Our goal was to quantify how much new semantic content is introduced throughout the optimization, which is important for understanding the evolution of prompts. To that end, our diversity score is a custom metric designed to capture two aspects:
> * Semantic richness, measured by counting the number of unique lemmas of content words (nouns, verbs, adjectives) after stop-word removal;
> * Uniqueness across prompts, defined as the number of words that appear only in a specific prompt and not in others.
>
> Both counts are normalized to form the final diversity score. While standard metrics such as lexical diversity exist, they do not directly capture inter-prompt uniqueness, which is central to our goal of analyzing how the prompts diversify during optimization.
>
> ****
> **6. Comparing existing work**
> > **Reviewer’s concern**: A major concern is the absence of comparison of results with existing work. The comparisons on variants of experiments (although logical) are crafted in this paper itself.
>
> **Our Response**:
> We thank the reviewer for this observation. Our paper addresses **a novel problem setting**--self-enhancing prompt optimization for fine-grained image classification without labels using MLLMs--which, to the best of our knowledge, has **not been thoroughly explored in prior work**. As a result, direct comparisons with existing methods are limited.
>
> We try to find some related methods. We applied the **SPO**[4] method from the text domain in our setting, whose results are reported in paper Table 1. We additionally applied the **Many-shot In-Context Learning (ICL)**[6] method to our setting by adding unlabeled images directly to the context window for prediction. We also add another baseline according to reviewer Eaom to use the MLLM to label some unlabeled data as demonstration examples. The results in the Table below show that our method outperforms these baselines in most of the settings.
>
> **Table: Results of new baselines.**
> | CUB_cuckoo  | Gemini 1.5 Flash | GPT-4o  | CUB_oriole | Gemini 1.5 Flash | GPT-4o   |
> |:-|:-|:-|:-|:-|:-|
> | Few-shot with MLLM labels | 53.41 ± 5.31 | 61.22 ± 2.82 | Few-shot with MLLM labels | 57.75 ± 4.42     | 64.94 ± 7.96 |
> | Many-Shot ICL [6]  | 48.05 ± 6.84     | 52.68 ± 9.27  | Many-Shot ICL [6]  | **59.78** ± 6.60 | 71.46 ± 8.75 |
> | AutoSEP (ours)  | **61.22** ± 2.30 | **75.12** ± 1.83  | AutoSEP (ours)  | 58.20 ± 0.88  | **75.96** ± 0.90 |
> | **CUB_vireo**  |  |   | **iNat_butterfly**  |
> | Few-shot with MLLM labels | 44.94 ± 8.56     | 72.36 ± 10.5      |Few-shot with MLLM labels  | 51.43 ± 3.26  | 82.57 ± 14.3 |
> | Many-Shot ICL [6]   | 56.18 ± 7.95     | 68.54 ± 13.6      |Many-Shot ICL [6]  | 55.71 ± 1.56     | 72.29 ± 12.2 |
> | AutoSEP (ours)  | **59.18** ± 0.53 | **87.42** ± 1.10  |AutoSEP (ours) | **66.57** ± 1.42 | **82.71** ± 2.43 |
> | **iNat_lupine**   |  |   |**StanfordDogs_terrier**    |
> | Few-shot with MLLM labels | 60.22 ± 11.4 | 69.33 ± 3.89  |Few-shot with MLLM labels  | 66.67 ± 10.3     | 87.56 ± 3.47 |
> | Many-Shot ICL [6]  | 63.47 ± 7.71  | 74.00 ± 5.54      |Many-Shot ICL [6]  | 65.87 ± 4.04     | 79.78 ± 9.04 |
> | AutoSEP (ours)  | **66.15** ± 1.47 | **76.60** ± 1.40  |AutoSEP (ours)  | **69.28** ± 0.97 | **92.18** ± 1.09 |
>
> We believe these efforts provide a fair and informative benchmark, demonstrating the effectiveness and novelty of our method. We will add them to our revised paper.
>
> **Reference:**
>
> [4] Xiang, Jinyu, et al. "Self-supervised prompt optimization." arXiv preprint arXiv:2502.06855 (2025).
>
> [6] Jiang, Yixing, et al. "Many-shot in-context learning in multimodal foundation models." arXiv preprint arXiv:2405.09798 (2024).

---

> > ### Author Response · Authors · 2025-08-06
> >
> > Dear reviewer,
> >
> > We really appreciate your detailed and constructive comments. We have carefully addressed each of your questions in our rebuttal. Specifically:
> > - We demonstrated that including or excluding parent class names in the initial prompt has minimal impact on final performance.
> > - We justified the computational complexity of our method, highlighting its modest runtime relative to the consistent and substantial performance gains.
> > - We clarified our benchmark subset creation, grounding it in both empirical evidence and prior literature.
> > - We included additional experiments using a smaller MLLM (Qwen2 VL 7B), which still achieved substantial performance gains.
> > - We clarified the definition of our diversity score.
> > - We updated baseline comparisons and explained the use of certain existing method comparisons.
> >
> > We hope our response has addressed your concerns satisfactorily. If you have any remaining questions or suggestions, we would be happy to further clarify. Thank you again for your time and feedback.
> >
> > Sincerely,
> >
> > Anonymous Authors

---

> > ### Comment · Reviewer_AiFG · 2025-08-07
> >
> > The rebuttal substantially addresses my concerns. According to my understanding, it addresses the comments by other reviewers too. I appreciate this rebuttal and  am inclined to upgrade my rating.

---

> > > ### Author Response · Authors · 2025-08-08
> > >
> > > Thank you sincerely for your thoughtful response and for considering an upgrade to your score. We’re delighted to hear that our rebuttal addressed your concerns successfully. We truly appreciate your time and the valuable feedback you provided throughout the review process.

---

### Official Review · Reviewer_B2au · 2025-06-30

**Clarity:** 1
**Significance:** 2
**Originality:** 3
**Rating:** 4
**Confidence:** 4

**Summary:**

This paper proposes AutoSEP, an automatic self-enhancing prompt learning framework that iteratively refines the description prompt to generate more discriminative descriptions for fine-grained zero-shot image classification.

**Questions:**

1. Is this method entirely training-free, with all steps performed at test time? More specifically, does each image have its own prompt that is iteratively optimized to generate a more discriminative description? If so, since the prompt is continuously changing, would this introduce instability or inconsistency during inference?
2. The iterative prompt optimization appears to be potentially time-consuming for each image. How long does the inference process take in practice? It would be helpful if the authors could provide details on the inference pipeline and report the average inference time per image.

Other questions and concerns are outlined in the “Weaknesses” section above. If the authors can address these issues satisfactorily, I would be open to revising my score.

**Ethical Concerns:**

["NO or VERY MINOR ethics concerns only"]

**Final Justification:**

The supplementary explanations and ablation results have effectively addressed the concerns I raised earlier.r. Given these improvements, I am inclined to update my overall score accordingly.

**Limitations:**

Yes, but the paper should more explicitly address the limitations, as outlined in the Weaknesses and Questions.

**Quality:**

3

**Strengths And Weaknesses:**

Strengths:
1. The idea of self-enhancing prompt learning in a fully unsupervised setting is novel and effectively enriches instance-level semantic information.
2. The paper is well-structured and clearly written, making it easy to understand the methodology and contributions.
3. The experiments are comprehensive and demonstrate the effectiveness of the proposed framework across multiple fine-grained benchmarks.

Weakness:
1. Equation (3) is unclear. It is not obvious how the random variable $Z$ affects the evaluation of $V(x_i, x_j)$, since $Z$ is not explicitly included in the function's notation. This could be misleading to readers unfamiliar with the design.
2. The definition of $G(t_a, t_b)$ is ambiguous. It would be helpful to clarify what $G$ represents and what its output is — for example, whether it is a textual prompt template or a more structured representation.
3. It would be helpful to clarify the function of $M$ and what $M(x_i, G(t_i, t_j)) = \text{``first''}$ actually means. Does ``first'' always refer to $t_i$, or is the order randomized depending on $Z$? Without clarification, it is difficult to interpret how correctness is determined and how $V(x_i, x_j)$ reflects model performance.
4. During the sampling process in Eq.~(3), what happens if the model encounters visually similar images from the same class? Such cases may introduce ambiguity and make it difficult for the model to consistently distinguish between the corresponding descriptions, potentially affecting the reliability of the evaluation.
5. In Section 3.3.2, it is unclear how to get $J_{error}$. How to identify that the generated descriptions failed to correctly distinguish between images? Is it manually defined?

---

> ### Author Rebuttal · Authors · 2025-07-30
>
> We thank the reviewer for their thorough review and feedback. We’ll answer the questions one by one.
>
> **1, 2, 3, & 5. Regarding algorithm notations**
>
> **Our Response**:
> We thank the reviewer for the detailed comments on the mathematical notations. We clarify each point regarding algorithm notations in detail below.
>
> * **Clarification of $V(x_i, x_j)$**: $V(x_i, x_j)$ is a binary correctness indicator that reflects whether the MLLM can correctly match image $x_i$ to its corresponding description $t_i$ over an incorrect alternative $t_j$. While the variable $Z \sim \text{Bernoulli}(0.5)$ is not explicitly included in the notation, it determines the random ordering of the two descriptions in the binary-choice prompt and thus influences the evaluation.
> * **Definition of $G(t_a, t_b)$**: $G(t_a, t_b)$ is a prompt template that formats two descriptions $t_a$ and $t_b$ into a binary-choice question posed to the MLLM, asking the MLLM whether the first description $t_a$ or the second description $t_b$ correctly describes the image. The MLLM is then expected to answer with either “first” or “second”.
> * **Role of $M$ and interpretation of $M(x_i, G(t_i, t_j)) = \text{“first”}$**: We define the function of $M$ in Sec. 3.1, whose input is an image and a prompt, and output is the MLLM response. Specifically, $M(x_i, G(t_i, t_j))$ is the MLLM response (either "first" or "second") when shown image $x_i$ and prompt $G(t_i, t_j)$. To reduce positional bias, the order of $t_i$ and $t_j$ is randomly determined using $Z \sim \text{Bernoulli}(0.5)$. If $Z=0$, we prompt with $G(t_i, t_j)$, and the correct answer is “first”; if $Z=1$, we prompt with $G(t_j, t_i)$, and the correct answer is “second”. We use an indicator function to return $1$ if the model selects the correct description (i.e., matches the image to its ground-truth description), and 0 otherwise.
> * **Clarification of $J_{error}$**: $J_{error}$ denotes the set of image pairs where the MLLM fails to choose the correct description. That is, for a given pair $(x_i, x_j)$, if $V(x_i, x_j) = 0$, then $(x_i, x_j)\in J_{error}$.
>
>
> ****
> **4. Concern of sampling process**
>
> > **Reviewer’s concern**: During the sampling process in Eq.~(3), what happens if the model encounters visually similar images from the same class? Such cases may introduce ambiguity and make it difficult for the model to consistently distinguish between the corresponding descriptions, potentially affecting the reliability of the evaluation.
>
> **Our Response**:
> The goal of our description generation process is to **elicit detailed, fine-grained understanding of each image**, instead of enforcing class-level discrimination. This image-specific descriptive richness is then used to guide the classification process in a separate classification prompt. In this context, encountering visually similar images--even from the same class--is not a flaw but an opportunity: it encourages the model to uncover subtle, discriminative visual cues that may otherwise be overlooked.
> In traditional self-supervised learning methods, similar images from the same class are often sampled together to learn meaningful feature representations through contrastively learning. Our framework is inspired by this idea and pushes the model to verbalize these subtle differences, which ultimately improves the downstream classification.
>
> Besides, we conducted ablation studies to see how the sampling process affects the final results. We conducted our AutoSEP with only sampling different-class images or only sampling same-class images for the instance-level classification. The results are in the Table below. Same-class and different-class sampling both don’t affect the final results much.
>
> **Table: Results of different sampling processes.**
> | CUB_cuckoo                                     | Gemini 1.5 Flash |
> |:------------- |:-------------:|
> | AutoSEP with different-class image sampling    | 60.98 ± 3.59     |
> | AutoSEP with same-class image sampling         | 61.58 ± 3.11     |
> | AutoSEP with random image sampling (ours)      | 61.22 ± 2.30     |
>
>
> ****
> **5. Addressing reviewer’s questions**
>
> > **Q1**: Is this method entirely training-free, with all steps performed at test time? More specifically, does each image have its own prompt that is iteratively optimized to generate a more discriminative description? If so, since the prompt is continuously changing, would this introduce instability or inconsistency during inference?
>
> **Answer 1**: To clarify, our method is **entirely training-free** in the sense that it does not require any finetuning of the model parameters. We only require black-box access to the MLLM, and perform optimization externally through prompt. **The prompt is not optimized per image.** Instead, we iteratively optimize a single description prompt using a small set of unlabeled images. This optimized prompt is then used consistently across all test-time images to generate improved image descriptions.
>
> > **Q2**: The iterative prompt optimization appears to be potentially time-consuming for each image. How long does the inference process take in practice? It would be helpful if the authors could provide details on the inference pipeline and report the average inference time per image.
>
> **Answer 2**: One optimization iteration on 60 unlabeled images using Gemini 1.5 Flash takes approximately 12 minutes on average, as reported in Sec. 4.3. This iterative prompt optimization is performed **once offline**, not per image. After the prompt is optimized, **inference is highly efficient**--each image requires only a single forward pass for description generation and one for prediction. As a result, the average inference time per image is comparable to standard zero-shot inference, with negligible additional overhead introduced by our method.

---

> > ### Comment · Reviewer_B2au · 2025-08-05
> >
> > I appreciate the authors’ thorough and thoughtful rebuttal. The supplementary explanations and ablation results have effectively addressed the concerns I raised earlier. I recommend incorporating these clarifications into the main paper. Given these improvements, I am inclined to update my overall score accordingly.

---

> > > ### Author Response · Authors · 2025-08-06
> > >
> > > We sincerely thank you for your detailed feedback and we’re very happy to hear that our rebuttal effectively addressed your concerns. We’ll definitely incorporate the clarifications and ablation results into our main paper as you suggested. Thank you again for your time and effort in reviewing our paper and providing insightful feedbacks!

---

### Official Review · Reviewer_Eaom · 2025-07-02

**Clarity:** 3
**Significance:** 2
**Originality:** 2
**Rating:** 4
**Confidence:** 4

**Summary:**

The paper presents AutoSEP, a training-free approach to enhance MLLLM’s fine-grained image classification performance. The key idea is to use some unlabeled data to optimize the description prompt for classification. Empirical evaluation demonstrates performance improvement on multiple fine-grained classification datasets.

**Questions:**

1. \[Line 106\] The equation there should be numbered as well like (2).
2. \[Line 128\] Is \<IMG\> missing from the prompt?
3. The second question also applies to Figure 2
4. It’s better to specify the Gemini version (1.5 Flash) in Table 1\.
5. Is there any data resampling or are all datasets balanced in class distribution? If not, the use of accuracy should be justified. The description in Section 4.1 is insufficient.
6. \[Line 274\] It’s great you address the perspective of computational complexity. It’s useful to comment on the compute for baselines as well.

**Ethical Concerns:**

["NO or VERY MINOR ethics concerns only"]

**Final Justification:**

I've reviewed the rebuttal and comments from other reviewers. The two baselines added are convincing, so I've raised the score.

**Limitations:**

See Weakness 4

**Quality:**

3

**Strengths And Weaknesses:**

1. The problem setup of improving fine-grained image classification performance with black-box only access to MLLM is practical and useful.
2. It’s commonly observed that MLLM tend to under-perform for fine-grained classification, so the focus on fine-grained classification is also well motivated.
3. Leveraging the idea of contrastive learning for prompt optimization seems good.

Weaknesses

1. The focus on zero-shot classification is not well motivated given we typically have access to a small number of demonstration examples.
2. Given we have access to unlabeled data, it’s important to add a semi-supervised method as baseline as well. For example, we can use the MLLM to label those unlabeled data and use those cases as demonstration examples. The current baseline (4) only uses random labels.
3. \[Line 57\] It’s important to acknowledge there’s some previous work such as [https://arxiv.org/abs/2405.09798](https://arxiv.org/abs/2405.09798) showing adding unlabeled images directly to the context window boosts zero-shot classification performance. It’s important to add that as a baseline besides vanilla zero-shot. It’s similar to baseline (5) but without the prompt “"The first m images show distinct types of {specie}."”
4. The possibility of data leakage is not discussed, especially given most datasets used are old.

---

> ### Author Rebuttal · Authors · 2025-07-30
>
> We really appreciate the reviewer for their detailed and valuable feedback and constructive suggestions. We address all of the concerns below.
>
> **1. The motivation of using limited unlabeled data**
>
> > **Reviewer’s concern**: The focus on zero-shot classification is not well motivated given we typically have access to a small number of demonstration examples.
>
> **Our Response**:
> We appreciate the reviewer’s comment and would like to clarify our motivation and contribution. While it is true that a few labeled examples may sometimes be available, this is **not guaranteed** in many real-world scenarios, particularly in fine-grained domains (e.g., biodiversity, rare diseases, or emerging scientific categories), where **labeled data is expensive or infeasible to obtain**. Our method specifically addresses the *fully unsupervised setting*, where no labeled data is available, but a small number of unlabeled images can be collected. This setting is practically relevant and underexplored in the context of MLLMs.
>
> For example, in nature science, researchers often collect unlabeled wildlife photos from camera traps but lack taxonomic annotations for rare species. Our method can be perfectly used in this case to increase the MLLM fine-grained image understanding and prediction precision.
>
>
> ****
> **2 & 3. Adding new baselines**
>
> > **Reviewer’s concern**:
> > * Given we have access to unlabeled data, it’s important to add a semi-supervised method as baseline as well. For example, we can use the MLLM to label those unlabeled data and use those cases as demonstration examples. The current baseline (4) only uses random labels.
> > * It’s important to acknowledge there’s some previous work showing adding unlabeled images directly to the context window boosts zero-shot classification performance. It’s similar to baseline (5) but without the prompt "The first m images show distinct types of {specie}."
>
> **Our Response**:
> We sincerely thank the reviewer for their insightful suggestions regarding two new baseline comparisons. We list the results of the newly added baselines in the Table below.
>
> **Table: Results of new baselines.**
> | CUB_cuckoo      | Gemini 1.5 Flash | GPT-4o       | CUB_oriole      | Gemini 1.5 Flash | GPT-4o       |
> |:-------------|:-------------|:-------------|:-------------|:-------------|:-------------|
> | Few-shot with MLLM labels | 53.41 ± 5.31     | 61.22 ± 2.82      | Few-shot with MLLM labels | 57.75 ± 4.42     | 64.94 ± 7.96 |
> | Many-Shot ICL [1]         | 48.05 ± 6.84     | 52.68 ± 9.27      | Many-Shot ICL [1]         | **59.78** ± 6.60 | 71.46 ± 8.75 |
> | AutoSEP (ours)            | **61.22** ± 2.30 | **75.12** ± 1.83  | AutoSEP (ours)            | 58.20 ± 0.88     | **75.96** ± 0.90 |
> | **CUB_vireo**             |  |               | **iNat_butterfly**      |
> | Few-shot with MLLM labels | 44.94 ± 8.56     | 72.36 ± 10.5      |Few-shot with MLLM labels  | 51.43 ± 3.26     | 82.57 ± 14.3 |
> | Many-Shot ICL [1]         | 56.18 ± 7.95     | 68.54 ± 13.6      |Many-Shot ICL [1]          | 55.71 ± 1.56     | 72.29 ± 12.2 |
> | AutoSEP (ours)            | **59.18** ± 0.53 | **87.42** ± 1.10  |AutoSEP (ours)             | **66.57** ± 1.42 | **82.71** ± 2.43 |
> | **iNat_lupine**           |  |               |**StanfordDogs_terrier**      |
> | Few-shot with MLLM labels | 60.22 ± 11.4     | 69.33 ± 3.89      |Few-shot with MLLM labels  | 66.67 ± 10.3     | 87.56 ± 3.47 |
> | Many-Shot ICL [1]         | 63.47 ± 7.71     | 74.00 ± 5.54      |Many-Shot ICL [1]          | 65.87 ± 4.04     | 79.78 ± 9.04 |
> | AutoSEP (ours)            | **66.15** ± 1.47 | **76.60** ± 1.40  |AutoSEP (ours)             | **69.28** ± 0.97 | **92.18** ± 1.09 |
>
> While these two baselines are stronger than baseline (4) and (5) in our original paper, our proposed method still **consistently outperforms** them across multiple experiment settings. This further demonstrates the advantage of our self-enhancing prompt learning framework.
>
> We will add these additional baselines and results to the revised paper to strengthen the effectiveness of our work.
>
>
> **Reference:**
>
> [1] Jiang, Yixing, et al. "Many-shot in-context learning in multimodal foundation models." arXiv preprint arXiv:2405.09798 (2024).
>
>
> ****
> **4. Adding new baselines**
>
> > **Reviewer’s concern**: The possibility of data leakage is not discussed, especially given most datasets used are old.
>
> **Our Response**:
> We appreciate the reviewer’s concern regarding the possibility of data leakage. In our experiment, we specifically selected fine-grained classification tasks where the **MLLM’s zero-shot performance is low** and unsatisfying. This underperformance suggests that the model can’t do well in recognizing these fine-grained images and subtle visual distinctions​​, indicating a lack of prior exposure to the specific label sets or task formulations. While we acknowledge that it’s possible to have data leakage in the MLLMs, and complete verification of data leakage is challenging. However, the poor baseline performance provides empirical evidence that the model has not memorized these tasks. Moreover, our work focuses on improving zero-shot performance using only unlabeled data, regardless of whether the model has seen related data before.
>
>
> ****
> **5. Addressing reviewer’s questions**
>
> **Q1 & Q4**:
> **Answer**: Thanks to the reviewer for pointing out these helpful suggestions. We will revise the paper to include the missing equation number for consistency and clarify the Gemini version (1.5 Flash) in Table 1.
>
> **Q2 & Q3**: [Line 128] Is `<IMG>` missing from the prompt?
> **Answer**: In our setup, we do send the image together with the prompt to the MLLM, but we do not explicitly include an `<IMG>` token in the text prompt. This is because models like Gemini 1.5 Flash handle multimodal inputs through dedicated APIs or structured inputs, where the image is passed as a separate input alongside the text, not embedded in the textual prompt.
>
> **Q5**: Is there any data resampling or are all datasets balanced in class distribution?
> **Answer**: All the results reported are evaluated on datasets that are **balanced** in class distribution. We will clarify this detail explicitly in Section 4.1 in the revised paper.
>
> **Q6**: [Line 274] It’s great you address the perspective of computational complexity. It’s useful to comment on the compute for baselines as well.
> **Answer**:
> For optimization-free baselines, the computational cost is relatively low, since they do not require iterative prompt refinement. For optimization-based baselines, we used the same experimental setup as our proposed method, resulting in comparable computational complexity. More specifically, optimization with random labels takes approximately $O(2(1+n)·bl)=O(40n)$ MLLM queries per iteration, as it involves only a single classification per image. Optimization with majority vote takes approximately $O(2(1+n)·bl+cn)=O(45n)$ queries per iteration, with $c=5$ denoting the number of majority votes collected for updates in each iteration. SPO is more lightweight, requiring only $O(2n)$ queries per iteration, since it only compares with the previous prompt and does not rely on several candidate prompts. We will include these details in the revised paper to provide a clearer understanding of the computational costs.

---

> > ### Comment · Reviewer_Eaom · 2025-08-01
> >
> > Thanks for the great rebuttal. I've raised my score

---

> > > ### Author Response · Authors · 2025-08-06
> > >
> > > Thank you so much for your response and raising your score. We really appreciate it! We’re glad to hear that you found our rebuttal satisfactory. Your thoughtful feedback has been invaluable in helping us improve our work. Thank you again for your time and support!

---

### Official Review · Reviewer_gnDg · 2025-07-02

**Clarity:** 4
**Significance:** 2
**Originality:** 3
**Rating:** 5
**Confidence:** 4

**Summary:**

This paper introduces AutoSEP, a method that aims to improve the zero-shot fine-grained classification ability of MLLM via automatic prompt optimization.  The proposed method requires the MLLM to generate a detailed image description and, based on that description, to conduct fine-grained classification. The core idea is to iteratively improve the description-generation prompt using unlabelled data.  To achieve this, it samples positive and negative image-text pairs (like contrastive learning) and asks MLLM to align the generated description with the correct image. Once fails, it asks the MLLM to refine the initial prompt. Experiments show the effectiveness of the proposed method.

**Questions:**

Since the method is simple and the paper is clear, I do not have any questions to clarify the paper.  It is noticeable that GPT-4o outperforms Gemini by a large margin across all benchmarks, while in my own experience, Gemini 2.0/2.5 performs much better than GPT on seeing image details.  I would appreciate it if you could verify this.  Please also see the weakness.

**Ethical Concerns:**

["NO or VERY MINOR ethics concerns only"]

**Final Justification:**

I will slightly raise my rating since some of my concerns have been resolved.

**Limitations:**

Yes

**Quality:**

3

**Strengths And Weaknesses:**

[Strength]
1. The paper is easy to follow and well organized.
2. The proposed method is logically clear and very simple.  It is an elegant combination of automatic prompt optimization with contrastive learning in fine-grained classification task, and I am happy to see that such a simple method actually brings consistent improvement across multiple models and benchmarks
3. The experiments are sufficient and convincing.

[Weakness]
Although it is an elegant combination of two methods, the novelty of this method is somewhat limited.  The core of this method is still automatic prompt optimization, which is not a new idea.  Fitting automatic prompt optimization to fine-grained image classification is a delightful attempt, but cannot be recognized as a significant creation.  The scope of fine-grained classification also limits the significance of the proposed method.

---

> ### Author Rebuttal · Authors · 2025-07-30
>
> We’re very grateful to the reviewer’s valuable feedback. Below is our response to their concerns.
>
> **1. On method novelty**
>
> > **Reviewer’s concern**: Although it is an elegant combination of two methods, the novelty of this method is somewhat limited. The core of this method is still automatic prompt optimization, which is not a new idea. Fitting automatic prompt optimization to fine-grained image classification is a delightful attempt, but cannot be recognized as a significant creation. The scope of fine-grained classification also limits the significance of the proposed method.
>
> **Our Response**:
> We appreciate the reviewer’s perspective and agree that automatic prompt optimization is not a new concept in itself. However, our contribution is not a reintroduction of this idea but rather a novel framework for its application in a new and challenging domain. Specifically, our contributions are:
>
> * To show **how unsupervised images can improve zero-shot prediction performance of LLMs**, a previously underexplored research direction. This setting is important for several real applications, such as medical image diagnosis where labeled data is scarce but unlabeled data are readily available.
> * To achieve this goal, since only unlabeled images are given, **traditional prompt optimization algorithms that rely on supervised data cannot be applied**. To resolve this issue, we propose a new framework to connect unsupervised prompt optimization with contrastive learning which uses **pairwise reasoning and self-supervised feedback** to iteratively improve descriptions that guide classification. This is the main technical contribution of the paper.
>
> Our work **directly addresses a persistent and underexplored weakness of current MLLMs**: their performance is limited in domains with insufficient or no training exposure. This issue continues to arise, as new tasks and data distributions emerge. There will always be scenarios where labeled training data is scarce or inaccessible due to cost, rarity, or privacy constraints. Existing methods provide no effective or practical solution for adapting MLLMs in such settings.
>
> Our work specifically targets these underperforming domains by enabling unsupervised prompt optimization without label reliance. Fine-grained classification serves as a prime example, but our method is **broadly applicable** to other datasets and tasks where the MLLM's training exposure is limited.
>
>
> **2. Results of better Gemini models**
>
> > **Reviewer’s concern**: It is noticeable that GPT-4o outperforms Gemini by a large margin across all benchmarks, while in my own experience, Gemini 2.0/2.5 performs much better than GPT on seeing image details. I would appreciate it if you could verify this.
>
> **Our Response**: We appreciate the reviewer’s comment and agree that Gemini 2.0/2.5 models have demonstrated strong image understanding capabilities in many scenarios. To further investigate this, we conducted additional experiments using Gemini 2.0 Flash, as shown in the Table below:
>
> **Table: Results on Gemini 2.0 Flash.**
> | CUB_vireo                   | Gemini 2.0 Flash   | StanfordDogs_terrier        | Gemini 2.0 Flash   |
> |:-------------|:-------------|:-------------|:-------------|
> | Vanilla zero-shot           | 64.27 ± 1.31       | Vanilla zero-shot           | 74.60 ± 0.47       |
> | Zero-shot with descriptions | 64.27 ± 1.65       | Zero-shot with descriptions | 74.12 ± 1.21       |
> | AutoSEP (ours)              | **69.05** ± 1.78   | AutoSEP (ours)              | **85.94** ± 1.05   |
>
> While we observe that Gemini 2.0 Flash significantly outperforms Gemini 1.5 Flash, our results do not reflect a performance advantage over GPT-4o on our benchmark tasks. Nonetheless, the key takeaway is that our AutoSEP method **consistently improves** the performance of Gemini 2.0 Flash across both datasets, reaffirming its effectiveness across different MLLMs. We will include these new results in our revised paper.

---

### Decision · Program_Chairs · 2025-09-17

**Decision:**

Accept (poster)

**Comment:**

The paper addresses the problem of fine-grained recognition in zero-shot setting for multimodal LLMs. The core idea is to leverage unlabelled data to iteratively optimize the description-generation prompt.
Some notable strengths of the paper include:
- easy to read and well-organized paper
- proposed method is logically clear and elegant combination of existing components
- motivation for improving MLLM zero-shot performance in fine-grained scenarios is clear
- the ability of the method to use instance-level classification signal

Some important concerns of the paper were:
- limited novelty as it appears a combination of existing methods with only scope of fine-grained classification improvement
- no mention of semi-supervised baselines as the method leverage unlabelled data
- initial prompts use parent class information providing unfair advantage
- results with smaller variants of MLLMs
- comparison with some existing relevant works

After the post-rebuttal discussion period, reviewers acknowledged that their major concerns have been resolved. These include explanation on novelty, results on semi-supervised baselines, results on different sampling processes and smaller variants of MLLMs. Therefore, the decision is to recommend the acceptance and authors are recommended to include important reviewer's comments in the camera-ready version.